# OpenReview forum: "Steer Like the LLM: Activation Steering that Mimics Prompting"
_ICML.cc/2026/Conference — ICML 2026 spotlight_

### Official Review · Reviewer_dbvj · 2026-02-21

**Soundness:** 4
**Presentation:** 3
**Significance:** 3
**Originality:** 3
**Overall Recommendation:** 4
**Confidence:** 2

**Summary:**

The paper introduces a steering technique that mimics prompt-steering by using position-specific steering coefficients instead of constant ones. The method is trained either by minimizing the Mean Squared Error (MSE) between the prompt-steered activations and the PSR-intervened activations, or by maximizing the log-likelihood of the steered response.

**Compliance With Llm Reviewing Policy:**

Affirmed.

**Final Justification:**

See rebuttal acknowledgement.

**Key Questions For Authors:**

- Do the authors expect a prompt instruction like like "Replace all instances of the word X with the word Y when X is preceded by the word Z." could be steered with PSR? My suspicion is this relatively simple instruction that works with prompt steering would not work with even PSR due to the fact that this is not something can be simply done by modifying the activations (even position-specific activations) and this only works with prompt steering because the model can attend back to the raw natural language instruction in the prompt. If the authors agree this feels like an area of weakness I feel like this along with a a category of prompts that may not be suitable for PSR should be listed in some kind of Limitations section.

**Limitations:**

yes

**Strengths And Weaknesses:**

Strengths:
- The motivation for the method is clear and the method is original.
- The authors empirical finding that the effectiveness of prompt-steering interventions varies substantially across token positions provides strong motivation for introducing a token-specific coefficient function.
- The authors tested multiple training objectives and intervention targets (S-PSR vs. A-PSR).

Weaknesses:
- Assumption 3.1 is a relatively strong assumption to make as it is likely not true that prompt steering interventions operate along a single direction although the simplification helps motivate the method.
- The evaluation scope is fairly limited and it's not clear if this method would work to mimic more complex, multi-faceted attributes/styles/instructions that could be steered through normal prompt-steering. That being said the method seems to work better than constant coefficients some category of attributes.

---

> ### Author Rebuttal · Authors · 2026-03-30
>
> Thank you for reviewing our paper!
>
> Before going over your comments in detail, we would like to inform you that we extended our evaluation to a third steering benchmark. We evaluated our methods on the Gemma-2B-L20 and Gemma-9B-L20 splits of AxBench [1], each assessing steering on 500 concepts. With this evaluation, we compare against more steering methods (including finetuning, LoRA, and strong activation steering methods). We use the same experimental setup (temperature, maximum number of generated tokens, etc.) as [1]. The Tables below summarize the results. We set a new SOTA.
>
> Table I: Steering scores (↑) on AxBench for rank-1, single layer interventions:
> | | DiffMean [1] | SAE [1] | ReFT-r1 [1] | Φ-SV-r=1 [2] | S-Const-LL | S-PSR-LL | S-Const-MSE | S-PSR-MSE |
> |---|:---|:---|:---|:---|:---|:---|:---|:---|
> | Gemma-2B-L20 (500 concepts) | 0.178 | 0.151 | 0.509 | 0.606 | 0.504 | **0.618** | 0.311 | 0.367 |
> | Gemma-9B-L20 (500 concepts) | 0.322 | 0.191 | 0.630 | 0.892 | 0.633 | 0.667 | **0.903** | **0.900** |
>
> Table II: Steering scores (↑) on AxBench for multi-rank and/or multi-layer methods:
> | | Prompting [1] | LoReFT [1] | SFT [1] | LoRA [1] | LoRA-RePS [2] | LoReFT-RePS [2] | HyperSteer [3] | A-Const-LL | A-PSR-LL | A-Const-MSE | A-PSR-MSE |
> |---|:---|:---|:---|:---|:---|:---|:---|:---|:---|:---|:---|
> | Gemma-2B-L20 (500 concepts) | 0.731 | 0.722 | 0.714 | 0.641 | 0.793 | 0.805 | 0.742 | 0.792 | 0.690 | 0.783 | **0.871** |
> | Gemma-9B-L20 (500 concepts) | 1.075 | 0.777 | – | 0.602 | 0.631 | 0.757 | 1.091 | 0.757 | 0.827 | 1.053 | **1.120** |
>
>
> > Assumption 3.1 is a relatively strong assumption to make as it is likely not true that prompt steering interventions operate along a single direction although the simplification helps motivate the method.
>
> We agree with this point and already acknowledge this in the paper (lines 150-153 and 356-367), but we will expand on it in the revision. In relation to this, you may be interested in reading the "Significance of the contributions" paragraph in our response to reviewer sQFB.
>
> > The evaluation scope is fairly limited and it's not clear if this method would work to mimic more complex, multi-faceted attributes/styles/instructions that could be steered through normal prompt-steering. That being said the method seems to work better than constant coefficients some category of attributes.
>
> As mentioned above, we extended our evaluation scope by evaluating on a third steering benchmark, on 1,000 different concepts.
>
> > Do the authors expect a prompt instruction like like "Replace all instances of the word X with the word Y when X is preceded by the word Z." could be steered with PSR? My suspicion is this relatively simple instruction that works with prompt steering would not work with even PSR due to the fact that this is not something can be simply done by modifying the activations (even position-specific activations) and this only works with prompt steering because the model can attend back to the raw natural language instruction in the prompt. If the authors agree this feels like an area of weakness I feel like this along with a a category of prompts that may not be suitable for PSR should be listed in some kind of Limitations section.
>
> We set up an experiment to test whether the PSR models we investigated can express "replace Z X by Y" rules: We created five concrete (X, Y, Z) triplets. For each, we created 2x40 input-output examples, for training and evaluation, respectively. We were careful to construct the dataset such that models that would always replace `X` by `Y` regardless of whether `Z` precedes `X`, would get no answers correct.
>
> Across the 5x40 test examples using Gemma-2-9b-it, we find that:
> - A-PSR with a loglikelihood objective gets 199/200 answers correct, but A-PSR with the mean-squared error objective only gets 88/200 correct.
> - S-PSR with a loglikelihood objective gets 119/200 answers correct, but S-PSR with mean-squared error yields no correct answers.
>
> We conclude that A-PSR is expressive enough to represent this type of rule, though not in a way that is faithful to the prompt steering mechanics.
>
> Following your suggestion, we will add a section which highlights that the rank-1 interventions we explored in this paper cannot represent everything prompt steering can. We believe that investigating different architectures (e.g., multi-rank PSR variants) is an interesting path for future research in this regard.
>
> [1] Wu, Z. et al., AxBench: Steering LLMs? Even Simple Baselines Outperform Sparse Autoencoders. ICML 2025 *spotlight*
>
> [2] Wu, Z. et al., Improved Representation Steering for Language Models. NeurIPS 2025 *spotlight*
>
> [3] Sun, et al., HyperSteer: Activation Steering at Scale with Hypernetworks. Arxiv 2025. https://arxiv.org/abs/2506.03292

---

> > ### Author Rebuttal · Reviewer_dbvj · 2026-04-01
> >
> > I thank the authors for the detailed response, I have bumped up the scores a bit, but will keep my recommendation level. I believe the technique is novel and interesting. This is a good work that is presented well, but the claims maybe a bit overstated and the real impact is low. Reading some of the other reviews, I feel the work needs more evaluations to be more convincing of the utility of this kind of steering relative to prompting. It seems to me that a thin self-distillation LoRA would do the same thing as a far simpler technique.

---

> > > ### Author Response · Authors · 2026-04-01
> > >
> > > We thank the reviewer for their constructive feedback and positive review.
> > >
> > > The reviewer is right to point out that our evaluations do not show that activation steering is superior to prompting in general. This is not a message we intended to convey, rather, we believe that activation steering is a promising area (for the benefits of activation steering we refer to our response to reviewer sQFB) and that our work makes valuable contributions to it. We will make the wording of our claims more precise and will clarify that the benchmarks used for evaluation do not cover all instruction-following use cases.
> > >
> > > The same comment applies to LoRA, but it is worth noting that our AxBench results (Tables I–II in our earlier response) show that our best configuration (A-PSR-MSE) outperforms LoRA with different training objectives on both Gemma-2B and Gemma-9B, achieving a new state-of-the-art. Results for other LoRA variants can be found in [2], in Table II we reported the best-performing LoRA result.
> > >
> > > Based on the insights and results of our paper, we believe that a LoRA self-distillation variant that mimics the prompt steering interventions rather than only the token outputs, would be an interesting line of further research.

---

### Official Review · Reviewer_VzM8 · 2026-03-08

**Soundness:** 3
**Presentation:** 3
**Significance:** 3
**Originality:** 4
**Overall Recommendation:** 5
**Confidence:** 4

**Summary:**

This paper presents PSR (Prompt Steering Replacement), a family of activation steering methods that learn to replicate the effect of prompt steering on internal activations. The authors show that prompt steering induces position-varying intervention strengths across token positions. PSR estimates this steering strength by training a probe to predict the coefficients from the activations. Experiments across multiple LLMs show consistent improvements over constant-coefficient baselines, and faithful intervention replication on sycophantic data.

**Compliance With Llm Reviewing Policy:**

Affirmed.

**Final Justification:**

My final recommendaiton is based on the convincing rebuttal from the authors.

**Key Questions For Authors:**

- Have you analyzed the alignment between the learned probe weights w_attr and the difference-in-means direction for each trait and layer?

-  How does the average intervention strength evolve across layers? Do certain layers systematically contribute more to the steering effect, and if so, do these align with the layers identified as most effective for single-layer steering?

**Limitations:**

nan

**Strengths And Weaknesses:**

Strengths
- The paper is clearly written and easy to follow
- The idea of analyzing if prompting can be seen as one-dimension steering is interesting and well executed
- The paper contains many experimental results enabling to better understand how activation steering works

Weaknesses
- The paper claims interpretability as a key advantage of PSR over prompt steering, but do not analyzes what the learned probe weights w_attr encode. In particular, the comparison with difference-in-means direction would be an interesting analysis to see if the probe aligns with the representation of the target attribute
- Evaluation could be improved: The paper evaluates at a single coherence threshold (TA@C80) which sounds arbitrary. Reporting integrated values over a fixed coherence range (e.g. 70–95) would better report the performance of the approaches tested.
- Missing baselines: The paper does not report and compare their approach to other methods doing adaptive steering moduling the steering coefficient, such as [1] and [2].

References

[1] Hedström, A., Amoukou, S. I., Bewley, T., Mishra, S., & Veloso, M. (2025, October). To Steer or Not to Steer? Mechanistic Error Reduction with Abstention for Language Models. In International Conference on Machine Learning (pp. 22924-22945). PMLR.

[2] Vogels, A., Wong, B., Choho, Y., Blangero, A., & Bhan, M. (2025). In-Distribution Steering: Balancing Control and Coherence in Language Model Generation. arXiv preprint arXiv:2510.13285.

---

> ### Author Rebuttal · Authors · 2026-03-30
>
> Thank you for reviewing our paper!
>
> Before addressing your comments in detail, we would like to highlight that we extended our evaluation to a third steering benchmark. We evaluated our methods on the Gemma-2B-L20 and Gemma-9B-L20 splits of AxBench, each assessing steering on 500 concepts. We set a new SOTA on this benchmark, outperforming prompting, supervised finetuning (SFT), LoRA variants, and strong activation steering baselines. For the results, we kindly refer to our response to reviewer dbvj.
>
> **Interpretability**
>
> We believe our work contributes to interpretability in different ways:
> - The PSR framework we introduced enables testing simple architectures as hypotheses of prompt steering behavior. In the paper, we show that on a subset of target concepts, the rank-1 A-PSR is a promising hypothesis. These architectures and learned parameters could be a starting point for uncovering fine-grained trait-encoding circuits.
> - Faithful models of successful prompt steering can be used to investigate patterns in the failure modes of prompt steering by comparing PSR activations with prompt activations on prompt steering error cases.
>
> > - [Q1] Have you analyzed the alignment between the learned probe weights w_attr and the difference-in-means direction for each trait and layer?
> > - [Q2]: How does the average intervention strength evolve across layers? Do certain layers systematically contribute more to the steering effect, and if so, do these align with the layers identified as most effective for single-layer steering?
>
> Thank you for these suggestions. We did additional analyses on the persona vectors benchmark to address your questions and will include them in our revision.
> - Q1: Cosine similarities between the steering vectors of difference-in-means (DiM) and the all-layer configurations (A-{PSR|Const}-{MSE|LL}) were close to zero for most method-layer combinations. This is consistent with our intuition that steering vectors in all-layer and single-layer methods encode different things (see lines 185-190, right column). We observed the highest similarity (0.33) with A-PSR-MSE one layer before the steering layer.  Alignment between DiM and the single-layer configurations was higher overall. The highest cosine similarity was with S-PSR-MSE (0.48).
> - Q2: For the Llama models, the steering contributions of individual layers were high up to the early-middle layers and then dropped. For Qwen, there was an oscillating downward trend. For 8/9 cases (3 models x 3 traits), the single-layer steering layer was the layer with the last high steering contribution, or the layer after the last high-contribution layer.
>
> > Evaluation could be improved: The paper evaluates at a single coherence threshold (TA@C80) which sounds arbitrary. Reporting integrated values over a fixed coherence range (e.g. 70–95) would better report the performance of the approaches tested.
>
> We would like to clarify the following:
> * We measure trait alignment at multiple coherence points. In addition to TA@C80 (following [2]), we report TA@Cp (trait alignment at the coherence of prompting), enabling a direct comparison with prompting that, to our knowledge, is novel in the activation steering literature. We refer to Section 5.4 for details.
> * For a more fine-grained perspective we refer to Figure 8, which plots coherence-trait alignment curves for different traits and models.
> * On IFEval, we evaluate one steering coefficient to allow a fair comparison with [3], but we report the coherence values.
> * On the newly added AxBench evaluations, we use the evaluation metric of the benchmark: the harmonic mean between trait concept (i.e., trait alignment), instruct, and fluency (i.e., coherence) scores.
>
> > Missing baselines: The paper does not report and compare their approach to other methods doing adaptive steering moduling the steering coefficient, such as [1] and [2].
>
> Thank you for bringing these recent works to our attention. We will include them in our discussion of prior art.
>
> By evaluating on AxBench as a third benchmark, we now compare against more activation steering baselines.
>
> To briefly position our work against [1] and [2], we note that:
>
> - [1] and [2] dynamically set steering coefficients such that the presence of the steering vector in the steered activations (i.e., the projection of the steering vector onto the steered activations) becomes uniform across token positions. In this respect, they are closely related to the method of Stolfo et al., which we compared against (see Table 2). Similar to [2], Stolfo et al. set steering coefficients with the aim to keep steered activations consistent with the distribution of the last-response-token activations from positive training examples.
> - Our work revealed that there are many activations where prompt steering barely intervenes (see Figure 2). Consequently, the interventions of [1], [2], and Stolfo et al. are less faithful to the mechanics of prompt steering.

---

> > ### Author Rebuttal · Reviewer_VzM8 · 2026-04-03
> >
> > Thank you for you structured rebuttal. I've increased my score. I find this approach innovative, and it opens up interesting avenues for better understanding the connection between steering and prompting.

---

> > > ### Author Response · Authors · 2026-04-04
> > >
> > > Thank you for your review and for confirming your concerns have been addressed. Your comments and suggestions helped improve our paper.

---

### Official Review · Reviewer_sQFB · 2026-03-12

**Soundness:** 2
**Presentation:** 3
**Significance:** 2
**Originality:** 3
**Overall Recommendation:** 5
**Confidence:** 3

**Summary:**

The paper proposes prompt steering replacement, an activation steering method which is trained to mimic the effects of applying a particular intervention to the prompt. In contrast to existing steering methods, the authors show that prompting causes token-specific changes in activations. Following this, they design PSR, which is trained to mimic the changes in activations produced by prompting, enabling token-and-layer-specific control. They evaluate PSR on two tasks, persona steering and IFEval, and find that PSR either gets close to or exceeds the steering power of prompted interventions alone, while improving over prior activation steering techniques.

**Compliance With Llm Reviewing Policy:**

Affirmed.

**Final Justification:**

The rebuttal convinced me more of the value of the overall class of activation steering methods that they work on, which helped me further understand the value of their contribution within that line of work. Therefore, I have changed my evaluation.

**Key Questions For Authors:**

1. Could the authors discuss the relation, if any, of their results to Bigelow et al. [1]?
2. I'm left feeling unsure how the proposed method is actually good for interpretability. We learned that prompting can be reasonably modeled as per-token-scaled per-layer intervention vectors. As an interpretability tool, what does this tell us? What downstream use-cases or improvements to language models does this enable?

[1] Bigelow, Eric, et al. "Belief dynamics reveal the dual nature of in-context learning and activation steering." arXiv preprint arXiv:2511.00617 (2025).

**Limitations:**

yes

**Strengths And Weaknesses:**

**Soundness**

In my assessment, the paper is largely clear and technically sound. I had no issues understanding the methods and I believe the experimental results support the claims made. I am however skeptical of the claims that the effectiveness of the PSR method is comparable to that of prompting due to one underlying detail. As far as I can tell, the PSRs are trained using the activations of only *successful* completions to prompt intervention – cases where the prompt intervention worked. As a result, comparing the derived steering vectors to baseline prompting feels somewhat unfair, as the prompting baseline is left to incur losses of incorrect completions, but if the activation steering method was perfect it would never do so. Yet, we still see that steering does not always recover the performance of prompting. Perhaps we believe that the underparameterization of the steering intervention remediates this concern, but I am a bit unsure about this. Can the authors comment? What would happen if we trained the steering vectors on all responses to the intervention?

**Presentation**

Presentation quality is good – I found the writing and figures to be clear.

**Significance**

I appreciate the paper's aim to study methods at the intersection of activation steering and prompting. I think understanding the relationship between in-context behavioral conditioning and changes in-weights / in-activations is an important direction which is worthy of further study. I also agree that steering methods are potentially more robust than prompting interventions, but I'd like to see a citation or experiment to support this claim. I think overall my assessment of the paper's significance is dampened by not getting a clear view of the "so what" of the work – why would a practitioner reach for this method over prompting, which is simple, or just fully training the model (or a LoRA adapter)? What value do we gain from this rank-1 intervention?
I also find the evaluation somewhat limited, which is maybe contributing to this for me – I'd be more confident in the results if the authors could show that PSR is useful beyond the narrow tasks of persona steering and instruction following.

**Originality**

I believe the paper is technically original, and the precise design of the method is well thought-out based on the findings of per-token variance in activations. However, I don't see the method as a large technical innovation in the space of interventions one can make to a language model. In my assessment, the work is well-contextualized relative to other works on activation steering, and is solidly original overall.

---

> ### Author Rebuttal · Authors · 2026-03-30
>
> Thank you for your review! We address your comments below.
>
> **PSR does not always recover the performance of prompt steering despite relying on examples of successful prompt steering**
>
> Our paper includes discussion in this regard on lines 356-367. For the instructions where PSR methods underperform, we saw that the MSE training loss remained high after training. This indicates that rank-1 interventions are not always sufficient to represent (successful) prompt steering.
>
> This does not invalidate the PSR framework, however. The fact that A-PSR-MSE can faithfully replicate prompt steering interventions for persona concepts using simple operations and three orders of magnitude fewer parameters is a strong indication that position-specific steering coefficients are fundamental to the mechanics of prompt steering. For instructions where rank-1 interventions are insufficient, we suggest the extension of the evaluated PSRs to multi-rank variants as an interesting avenue for future research.
>
> We also note that it is common for activation steering methods to rely on supervision during training. Because concept detection has been shown to be a much easier task than steering [1], we believe that assuming access to a judge does not pose a constraint in practice.
>
> **Evaluation scope and relation to finetuning, LoRA, ...**
>
> We extended our evaluation to a third steering benchmark: we evaluate on AxBench, on 1,000 different target concepts (please refer to the Tables in our response to reviewer dbvj). We set a new SOTA on this benchmark, outperforming prompting, supervised finetuning (SFT), LoRA variants, and recent activation steering methods.
>
> **Significance of the contributions**
>
> You raised important questions regarding our contributions, and in our revision we will better highlight and disentangle them. We summarize our contributions with clarifications:
> - We present a new framework for training activation steering models: Our formulation of prompt steering as activation steering with the idea to replicate it with simpler, interpretable models is novel ***and agnostic to the specific rank-1 architectures that we explored in the paper***.
> - We propose and evaluate a first theory under this framework - based on two explicit assumptions - that accurately models prompt steering *on a subset of steering instructions*. We acknowledge in our paper that these assumptions do not hold for every prompt instruction. However, **we believe our results and analyses suggest that location-specific steering coefficients are a likely ingredient of more general theories.** The rank-1 interventions we studied could be a stepping stone, much like how physics often progresses by developing more general theories that include earlier ones as special cases.
> - Even though we do not claim that rank-1 PSR models are universally optimal steering models, **our best configurations outperform prompting, finetuning, and strong activation steering baselines, with different models and different target concepts from established benchmarks.** The persona steering task is also of great practical importance (see the introduction of [3], for instance).
>
> **Benefits of activation steering over prompting and finetuning**
>
> Our work advances activation steering, an area of great interest in the ML community. For instance, [1] and [2] both study rank-1 steering methods and were presented as spotlight papers at ICML 2025 and NeurIPS 2025, respectively. Benefits of activation steering include: more fine-grained steering control, reduced risk of forgetting or removing instructions in long prompts (e.g., https://venturebeat.com/security/meta-rogue-ai-agent-confused-deputy-iam-identity-governance-matrix), and potentially greater robustness against adversarial attacks [4]. Thank you for pointing out the last claim was missing a reference, we will add it in our revision.
>
> **Relation with Bigelow et al.**
>
> Bigelow et al. frame prompting and activation steering as *complementary techniques* influencing the belief p(c|x) in a latent concept c given the prompt. Their theory applies only to constant activation steering as they interpret it as an update to the prior p(c), which is input-independent. Our work shows prompting can be *viewed as activation steering*, with PSRs having both constant (α) and input-dependent (λ) components. Both perspectives are compatible.
>
> **What interpretability insights could PSRs unlock?**
> Due to space constraints, we kindly refer to our response to reviewer VzM8.
>
> [1] Wu, Z. et al., AxBench: Steering LLMs? Even Simple Baselines Outperform Sparse Autoencoders. ICML 2025 *spotlight*
>
> [2] Wu, Z. et al., Improved Representation Steering for Language Models. NeurIPS 2025 *spotlight*
>
> [3] Chen, R., et al., Persona Vectors: Monitoring and Controlling Character Traits in Language Models, Arxiv 2025.
>
> [4] Wang, H., et al. Steering Away from Harm: An Adaptive Approach to Defending Vision Language Model Against Jailbreaks, CVPR 2025

---

> > ### Author Rebuttal · Reviewer_sQFB · 2026-04-04
> >
> > I appreciate the authors' response, which makes good points about the value of activation steering work and the novelty of this method in revealing the mechanics of behavioral modulation of LLMs. I additionally appreciate the comparison against additional methods on AxBench, and I will raise my score.

---

> > > ### Author Response · Authors · 2026-04-04
> > >
> > > Thank you for your review and for confirming your concerns have been addressed. Your questions and constructive feedback helped improve our paper.

---

### Decision · Program_Chairs · 2026-04-30

**Decision:**

Accept (spotlight)

**Comment:**

**Summary:** This paper introduces Prompt Steering Replacement (PSR), a family of activation steering methods that distill prompt steering behavior into position-specific, low-rank interventions on model activations. The key insight is that prompt steering induces token-position-varying activation changes, motivating learned steering coefficients rather than the constant coefficients used in prior work. Experiments on persona steering, instruction following, and AxBench demonstrate improvements over constant-coefficient baselines and performance competitive with or exceeding prompt steering.

**Strengths:** Clear writing and well-motivated approach. The framing of prompt steering as activation steering is original. Strong empirical results across three benchmarks, including new SOTA on AxBench over prompting, SFT, LoRA, and recent activation steering methods.

**Weaknesses:** The rank-1 assumption is restrictive and does not hold for all steering instructions. Interpretability claims could be better substantiated with deeper analysis.

**Recommendation:** Accept.